# Parameter-Efficient Complementary Expert Learning for Long-Tailed Visual Recognition

Lixiang Ru
Ant Group
Hangzhou, China
rulixiang.rlx@antgroup.com

Xin Guo
Ant Group
Hangzhou, China
bangzhu.gx@antgroup.com

Lei Yu
Ant Group
Hangzhou, China
dubai.yl@antgroup.com

Yingying Zhang
Ant Group
Hangzhou, China
qichu.zyy@antgroup.com

Jiangwei Lao
Ant Group
Hangzhou, China
wenshuo.ljw@antgroup.com

Jian Wang
Ant Group
Hangzhou, China
bobblair.wj@antgroup.com

Jingdong Chen
Ant Group
Hangzhou, China
jingdongchen.cjd@antgroup.com

Yansheng Li
Wuhan University
Wuhan, China
yansheng.li@whu.edu.cn

Ming Yang
Ant Group
Hangzhou, China
m.yang@antgroup.com

## Abstract

Long-tailed recognition (LTR) aims to learn balanced models from extremely unbalanced training data. Fine-tuning pretrained foundation models has recently emerged as a promising research direction for LTR. However, we observe that the fine-tuning process tends to degrade the intrinsic representation capability of pretrained models and lead to model bias towards certain classes, thereby hindering the overall recognition performance. To unleash the intrinsic representation capability of pretrained foundation models, in this work, we propose a new Parameter-Efficient Complementary Expert Learning (PECEL) for LTR. Specifically, PECEL consists of multiple experts, where individual experts are trained via Parameter-Efficient Fine-Tuning (PEFT) and encouraged to learn different expertise on complementary sub-categories via a new sample-aware logit adjustment loss. By aggregating the predictions of different experts, PECEL effectively achieves a balanced performance on long-tailed classes. Nevertheless, learning multiple experts generally introduces extra trainable parameters. To ensure parameter efficiency, we further propose a parameter sharing strategy which decomposes and shares the parameters in each expert. Extensive experiments on 4 LTR benchmarks show that the proposed PECEL can effectively learn multiple complementary experts without increasing the trainable parameters and achieve new state-of-the-art performance.

## CCS Concepts

• **Computing methodologies** → **Computer vision tasks**.

## Keywords

Long-tailed Recognition, Parameter-Efficient Fine-Tuning, Multiple Experts

**ACM Reference Format:**

Lixiang Ru, Xin Guo, Lei Yu, Yingying Zhang, Jiangwei Lao, Jian Wang, Jingdong Chen, Yansheng Li, and Ming Yang. 2024. Parameter-Efficient Complementary Expert Learning for Long-Tailed Visual Recognition. In *Proceedings of the 32nd ACM International Conference on Multimedia (MM '24), October 28–November 1, 2024, Melbourne, VIC, Australia.* ACM, New York, NY, USA, 10 pages. https://doi.org/10.1145/3664647.3680799

## 1 Introduction

Real-world data usually conforms to a long-tailed distribution, where a minority of classes (*i.e.,* head classes) cover the majority of samples, while the other classes (*i.e.,* tail classes) comprise only a few samples. Such data imbalance poses a formidable challenge for learning long-tailed recognition (LTR) models, since the learned models are prone to be biased towards the head classes with most of data but exhibit poor performance on other classes. To address this challenge, researchers have proposed many approaches [17, 23, 36, 41, 51, 54] to learn competent models from unbalanced long-tailed training data.

Existing LTR approaches can be broadly categorized into *training from scratch* methods and *fine-tuning* methods. Training from scratch models are trained from randomly initialized models, and usually exhibit unsatisfactory performance [6, 55]. In contrast, LTR methods via fine-tuning [10, 16, 34, 43, 46] mitigate the data imbalance issue by adapting pretrained foundation models [24, 38, 40]. Foundation models are typically trained on extensive data, and inherently learn generalizable representations across different semantic categories [24, 40]. Therefore, LTR methods via fine-tuning foundation models often yield higher accuracies [34, 46]. For example, BALLAD [34], VL-LTR [46] and UDCPG [16] fine-tune the visual and linguistic branches of CLIP [40] for LTR. To accelerate the training process and prevent undermining the representation of pretrained models in full fine-tuning, LPT [10] and LIFT [43]

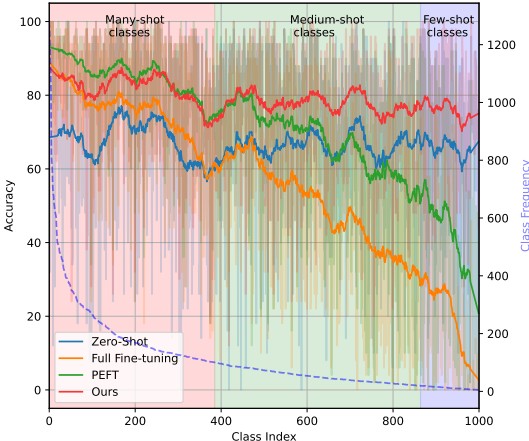

**Figure 1: Accuracy of each class in ImageNet-LT [9] by Zero-Shot recognition, Full Fine-tuning, PEFT (AdaptFormer [5]) and our proposed method. Here we use the pretrained CLIP-ViT-B [40]. For better visualization, the accuracy values are smoothed. Best viewed in color.**

introduce Parameter-Efficient Fine-Tuning (PEFT) methods to LTR and achieve notable improvements.

However, we empirically observe that full fine-tuning and PEFT on long-tailed data tend to degrade the inherent representation capability of pretrained models. To verify this, in Figure 1, we present the accuracy values of each class in ImageNet-LT [9], which are obtained by zero-shot recognition, full fine-tuning, PEFT (Adapt-Former [5]) and our proposed method, respectively. The class indices are sorted according to the class frequencies. Figure 1 shows that zero-shot recognition can achieve a relatively balanced performance on different class groups, *i.e.,* many-shot (with > 100 samples), medium-shot (with $20-100$ samples) and few-shot classes (with < 20 samples), suggesting the pretrained model learns satisfactory representations for different classes. Though full fine-tuning can achieve favorable accuracy on the many-shot classes, the performance on the medium and few-shot classes is inferior to zero-shot recognition, indicating the representation capability on these classes is degraded due to fine-tuning on long-tailed data. Similarly, PEFT exhibits a lower accuracy than zero-shot recognition on the few-shot classes, which indicates the representations of these classes are also hindered and become less discriminative.

To unleash the potential of pretrained foundation models, in this work, we propose the Parameter-Efficient Complementary Expert Learning (PECEL) for LTR. Specifically, PECEL learns multiple expert models from the pretrained models via PEFT, which are encouraged to attain different expertise on complementary classes. To ensure this, we propose a sample-aware logit adjustment (SLA) loss, which adjusts the learning difficulty of individual experts on different classes and samples. Thus, an expert only needs to focus on learning certain classes and pay less attention to other classes. By aggregating the predictions of these experts, PECEL can achieve a balanced performance across different classes. Besides, directly learning multiple experts generally increases the number of trainable parameters by multiple times. To alleviate this, we propose a parameter sharing strategy. For each expert, the parameters in different PEFT blocks are decomposed as shared components

and low-dimensional separate components. By distributing the shared components across different blocks, each expert only needs to maintain a small number of trainable parameters. To demonstrate the efficacy of PECEL, we conduct extensive experiments on the CIFAR100-LT [26], ImageNet-LT [9], Places-LT [57] and iNaturalist 2018 [47]. As shown in Figure 2, our proposed PECEL can perform on par with the state-of-the-art methods with notably fewer trainable parameters, and achieve higher accuracy with equivalent numbers of parameters.

To summarize, our contributions in this work include the following aspects.

- To the best of our knowledge, the proposed PECEL is the first work to effectively address the representation degradation issue when adapting foundation models for long-tailed recognition.
- We propose sample-aware logit adjustment and a parameter sharing strategy, which can effectively learn multiple complementary experts and reduce the number of trainable parameters, respectively.
- The proposed PECEL achieves state-of-the-art performance on 4 long-tailed recognition benchmarks.

## 2 Related Work

### 2.1 Long-Tailed Recognition

Long-tailed recognition (LTR) aims at learning models with balanced performance on different classes from extremely unbalanced training data. Existing LTR methods can be divided into training from scratch and fine-tuning pretrained models.

**Training from Scratch**. LTR methods by training from scratch mainly included re-balancing, logit adjustment, representation learning and ensemble learning. Re-balancing methods tackled the data imbalance issue via re-sampling the imbalanced inputs [31, 39, 42] or balancing the loss of different classes with re-weighting [6, 8, 13, 41] and re-margining [4]. Logit adjustment methods [28, 36, 56] sought to alleviate the class imbalance by incorporating dedicated biases or weights into the output logits. Representation learning methods [7, 22, 37, 58] aimed at learning balanced and discriminative representations from the long-tailed data via metric learning [35, 50, 53] or contrastive learning [18, 23, 29, 58]. Ensemble learning-based LTR methods [1, 45, 51, 54, 55] usually learned multiple diverse expert models, and aggregated the outputs of different experts. Specifically, RIDE [51] proposed a routing module and selected the most appropriate expert in testing time. SADE [54] trained complementary experts and adaptively aggregated the predictions to handle different data distributions. MDCS [55] and LGLA [45] aggregated skill-diverse experts to achieve more balanced performance. Compared to recent LTR methods that fine-tuned pretrained models [16, 43, 46], training from scratch methods usually exhibited inferior performance.

**Fine-tuning Pretrained Models**. Enlightened by the unprecedented success of foundation models [24, 38, 40], recent LTR methods [16, 34, 46] also resorted to pretrained foundation models to tackle the data imbalance. For example, BALLAD [34] trained the visual and linguistic encoder of CLIP [40] with text prompts and input images, then fine-tuned a linear adapter on the balanced training samples. Tian *et al.* proposed VL-LTR [46], which aligned the images

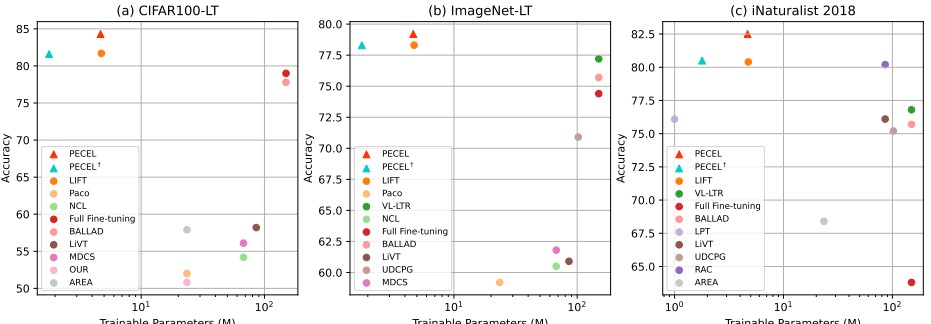

**Figure 2: Classification accuracy and number of trainable parameters of different methods on (a) CIFAR100-LT, (b) ImageNet-LT, (c) iNaturalist 2018. PECEL$^{\dagger}$ denotes PECEL with a lower bottleneck dimension.**

and text embeddings via contrastive learning, and trained the LTR classifier with cross-modality attention. GML [44] tackled LTR as a mutual information maximization problem [3] between the latent embeddings and ground-truth labels. UDCPG [16] addressed the feature collapse in training by aligning the image and text features with prototype-based contrastive learning. To improve the training efficiency and preserve the inherent representation capability [10, 43], some LTR methods also explored PEFT techniques for LTR. LPT [10] adopted two-stage visual prompt tuning [21] to further empower the pretrained model's capability on fine-grained discrimination. LIFT [43] proposed to integrate PEFT modules (such as AdaptFormer [5]) and designed a simple yet highly effective framework for LTR. As illustrated in Figure 1, we empirically observed that PEFT still degraded the representation capability of pretrained models. In this work, we propose to learn multiple complementary experts via PEFT to alleviate this issue and further unleash the potential of pretrained foundation models for LTR.

## 2.2 Parameter-Efficient Fine-Tuning

Parameter-efficient fine-tuning (PEFT) aims to relieve the training and storage cost of large pretrained models [38, 40] by only updating a small number of parameters while keeping other parameters frozen [30]. VPT [21] proposed to freeze the pretrained backbone and prepend some extra trainable prompt tokens to fine-tune visual recognition tasks. LoRA [20] proposed to inject low-rank trainable matrices to each layer in the backbone. To further explore parameter efficiency, VeRA [25] proposed to decompose the projection matrix in PEFT modules into a fixed random part and a learnable part. Different from LoRA-style methods that re-parameterized the projection weights, Adapter [19] inserted a bottleneck module after the feed-forward network (FFN) of Transformer blocks [48], which only comprised simple linear projections and nonlinear layers. AdaptFormer [5] extended the sequential design of Adapter to parallel and added an extra scaling factor, which demonstrated better performance. However, existing PEFT methods typically maintain separate parameters in each block, which will remarkably increase the number of trainable parameters when learning multiple experts for LTR. To address this issue, in this work, we propose a parameter sharing strategy by decomposing and sharing the parameters in different PEFT blocks.

## 3 Methodology

In this section, we first briefly introduce some preliminary knowledge, and then present the overall framework of the proposed Parameter-Efficient Complementary Expert Learning (PECEL). How to learn multiple complementary experts and reduce the number of trainable parameters in each expert model are elaborated in Section 3.3 and 3.4, respectively. Finally, the overall training objective and analysis on the number of trainable parameters of PECEL are illustrated in Section 3.5 and 3.6, respectively.

## 3.1 Preliminary

**Logit Adjustment**. Logit adjustment [36] tackles the class imbalance issue by infusing class-level biases (*i.e.,* relative class frequency) into the predicted logits in the training process. Let $\mathbf{z} \in \mathbb{R}^C$ denotes the model predicted logits, where $C$ denotes the number of classes, the logit adjustment loss is calculated as:

$$\mathcal{L}_{la}(\mathbf{z}, \mathbf{y}) = -\sum_{i=1}^{C} \mathbf{y}_i \log \frac{\exp(\mathbf{z}_i + \log(\mathbf{b}_i))}{\sum_{j=1}^{C} \exp(\mathbf{z}_j + \log(\mathbf{b}_j))}, \tag{1}$$

where $\mathbf{y} \in \mathbb{R}^C$ denotes the one-hot ground-truth label, $\mathbf{b}_i$ is the relative frequency of class $i$. Intuitively, Eq. 1 adds larger biases to the head classes, which can lower the learning difficulty, thereby enforcing the model to focus more on the tail classes and achieving more balanced performance.

**AdaptFormer**. The previous method LIFT [43] empirically demonstrates that conducting PEFT with AdaptFormer [5] can achieve favorable LTR performance. Therefore, in this work, we also adopt AdaptFormer as the basic PEFT block. Given the input feature $\mathbf{X} \in \mathbb{R}^{N \times D}$, where $N$ and $D$ denote the batch size and feature dimension, respectively, the output of an AdaptFormer block is

$$\begin{aligned} \mathbf{X}_{down} &= LN(\mathbf{X})\mathbf{W}_{down}, \\ \mathbf{X}_{up} &= ReLU(\mathbf{X}_{down})\mathbf{W}_{up}, \\ \mathbf{X}_{out} &= s \cdot \mathbf{X}_{up}, \end{aligned} \tag{2}$$

where $LN(\cdot)$ denotes Layer Normalization [2], $\mathbf{W}_{down} \in \mathbb{R}^{D \times d}$ and $\mathbf{W}_{up} \in \mathbb{R}^{d \times D}$ are the down and up projection matrix ($d$ is the bottleneck dimension and $d \ll D$), respectively, and $s$ is a learnable scalar. An AdaptFormer block is typically used as a parallel branch of the feed-forward network (FFN) in a Transformer block [11, 48], updating the outputs to $\mathbf{X}_{out} + FFN(\mathbf{X})$.

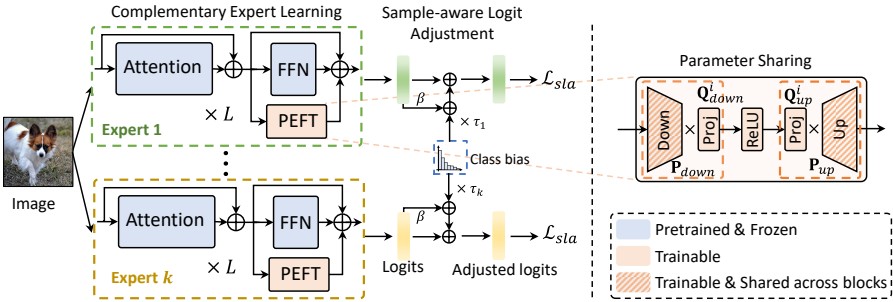

**Figure 3: Overview of the proposed Parameter-Efficient Complementary Expert Learning (PECEL). PECEL consists of multiple experts. In each expert, the CLIP-ViT-B[40] and AdaptFormer [5] with the proposed parameter sharing strategy are used as the pretrained model and basic PEFT block, respectively. To ensure different experts can learn complementary expertise, the experts are guided by the proposed sample-aware logit adjustment loss conditioned by different weight factors $\tau$. Besides, to ensure parameter efficiency when learning multiple experts, we propose a parameter sharing strategy to decompose and share the parameters in different PEFT blocks.**

## 3.2 Method Overview

The overall framework of the proposed Parameter-Efficient Complementary Expert Learning (PECEL) for LTR is illustrated in Figure 3. As shown in Figure 3, PECEL consists of multiple experts, where each expert is composed of a frozen pretrained backbone and trainable PEFT modules. In this work, following the previous works [43, 46], we also use the pretrained CLIP-ViT-B [40] and AdaptFormer [5] as the pretrained model and basic PEFT block in this work. To learn expertise on complementary classes, the output logits of different experts are supervised with the proposed sample-aware logit adjustment loss conditioned by different weighted factors. Since learning multiple experts introduces extra parameters, to ensure parameter efficiency, we propose a parameter sharing strategy to decompose and share the parameters in PEFT blocks. In the inference stage, the outputs of different experts are aggregated via averaging without adjusting the logits.

## 3.3 Learning Complementary Experts

As illustrated in Figure 1, the fine-tuning process undermines the inherent representation capability of pretrained models, then results in the model bias towards certain semantic classes. To take full advantage of pretrained models for LTR, we propose to learn multiple complementary experts, where each expert is enforced to learn the expertise on different subcategories via PEFT. The technical details of PEFT blocks in each expert are presented in Figure 3 and depicted in Section 3.4.

Unlike the fixed relative class frequency $\mathbf{b}$ in Eq. 1, to learn experts specializing in different class groups, we propose to condition $\mathbf{b}$ with different weight factors. Specifically, assuming the output logits of the $k-th$ expert $\mathbb{E}_k$ is $\mathbf{z}^k$, the loss function for $\mathbb{E}_k$ is calculated as:

$$\mathcal{L}_{la}(\mathbf{z}^k, \mathbf{y}) = -\sum_{i=1}^{C} \mathbf{y}_i \log \frac{\exp(\mathbf{z}_i^k + \tau_k \log(\mathbf{b}_i))}{\sum_{j=1}^{C} \exp(\mathbf{z}_j^k + \tau_k \log(\mathbf{b}_j))}, \quad (3)$$

where $\tau_k$ is a weight factor to enforce expert $\mathbb{E}_k$ to focus on specific classes. Particularly, when $\tau_k = 1$, Eq. 3 degrades to Eq. 1, and can facilitate a relatively balanced performance on different

classes. When $\tau_k > 1$, the imbalance ratio between different classes is enlarged, so the penalty to the head classes is increased, encouraging the expert $\mathbb{E}_k$ to focus and achieve better performance on tail classes. On the contrary, when $\tau_k < 1$, the imbalance ratio between different classes drops, therefore improving the performance on head classes for expert $\mathbb{E}_k$. By employing different weight factors, different experts can learn knowledge on complementary classes. In practice, we set PECEL with 3 experts to learn complementary knowledge on many-shot, medium-shot and few-shot classes by setting $\tau > 1$, $\tau = 1$ and $\tau < 1$, respectively.

Note that Eq. 3 only adjusts the learning difficulty according to the class frequency. However, due to the intra-class diversity, the learning difficulty of samples in the same class can be very different [56]. To address this issue and enforce the learning of each expert, in this section, we further adapt Eq. 3 and propose the Sample-aware Logit Adjustment (SLA). Specifically, SLA further adjusts the logit bias in Eq. 3 for the misclassified hard samples. Based on Eq. 3, the calculation of SLA loss is

$$\mathcal{L}_{sla}(\mathbf{z}^k, \mathbf{y}) = -\sum_{i=1}^{C} \mathbf{y}_i \log \frac{\exp(\mathbf{z}_i^k + \tau_k \log(\mathbf{b}_i) + \beta_i)}{\sum_{j=1}^{C} \exp(\mathbf{z}_j^k + \tau_k \log(\mathbf{b}_j) + \beta_j)}, \quad (4)$$

where $\beta \in \mathbb{R}^C$ is a bias vector and its values depend on the prediction results. Intuitively, if the current sample is correctly classified, $\beta = \mathbf{0}$, degrading Eq. 4 to Eq. 3. While the current sample is misclassified, $\beta_{i_p} = \epsilon$, $\beta_{i_y} = -\epsilon$ and $\beta_i = 0$ for other values, where $i_p$ and $i_y$ denote the index of the predicted and ground-truth class label, respectively. $\epsilon$ denotes a small positive value. For the misclassified samples, Eq. 4 can enlarge the logit bias on the ground-truth class, and decrease the bias on the misclassified class, thereby enforcing the model focus more on the misclassified samples.

## 3.4 Parameter-Efficient Expert Learning

Learning with Eq. 4 can yield multiple complementary experts. However, the number of trainable parameters will also increase linearly by multiple times. To ensure the parameter efficiency of PECEL, different from previous PEFT methods that maintain separate trainable parameters in different blocks [5, 20], we propose a

parameter sharing strategy to decompose and share the parameters across different PEFT modules.

**Parameter Sharing**. As illustrated in Figure 3, in each expert, the projection matrices in PEFT blocks are decomposed into shared base matrices and standalone low-dimensional matrices. Specifically, for the $i - th$ PEFT block, the down and up projection matrices are decomposed as:

$$\mathbf{W}^i_{down} = \mathbf{P}_{down}(\mathbf{Q}^i_{down})^\top,$$
$$\mathbf{W}^i_{up} = (\mathbf{Q}^i_{up})^\top \mathbf{P}_{up}, \tag{5}$$

where $\mathbf{P}_{down} \in \mathbb{R}^{D \times d}$ and $\mathbf{P}_{up} \in \mathbb{R}^{d \times D}$ denote the base down and up projection matrices, respectively. $\mathbf{P}_{down}$ and $\mathbf{P}_{up}$ are shared across different blocks. $\mathbf{Q}^i_{down} \in \mathbb{R}^{d \times d}$ and $\mathbf{Q}^i_{up} \in \mathbb{R}^{d \times d}$ are the standalone low-dimensional matrices for down projection and up projection in the $i - th$ block, respectively. Therefore, integrating Eq. 2, the outputs of the $i - th$ shared PEFT block is calculated as:

$$\mathbf{X}^i_{down} = LN(\mathbf{X}^i)(\mathbf{P}_{down}(\mathbf{Q}^i_{down})^\top),$$
$$\mathbf{X}^i_{up} = ReLU(\mathbf{X}^i_{down})((\mathbf{Q}^i_{up})^\top \mathbf{P}_{up}), \tag{6}$$
$$\mathbf{X}^i_{out} = s \cdot \mathbf{X}^i_{up}.$$

As shown in Figure 3, $\mathbf{X}^i_{out}$ and the outputs of pretrained backbone block are integrated via addition. Note that $\mathbf{Q}_{down}$ and $\mathbf{Q}_{up}$ are separate, the individual blocks can therefore maintain their own down and up projection matrices. Since $\mathbf{P}_{down}$ and $\mathbf{P}_{up}$ are shared, and given $d \ll D$, in each expert, the number of trainable parameters can be reduced by a large margin.

**Shared Parameter Regularization**. The reduction of trainable parameters requires the network to learn discriminative shared parameters. To facilitate this, we design a simple regularization loss for the shared parameters. Specifically, the shared parameters are restricted to orthogonal matrices. For the shared down projection matrix $\mathbf{P}_{down}$, the loss is

$$\mathcal{L}_{reg}(\mathbf{P}_{down}) = |\mathbf{P}^\top_{down}\mathbf{P}_{down} - \mathbf{I}|, \tag{7}$$

where $|\cdot|$ denotes the mean absolute error loss. $\mathbf{I}$ denotes the identity matrix. The calculation of $\mathcal{L}_{reg}(\mathbf{P}_{up})$ is in a similar way.

### 3.5 Training Objective

As shown in Figure 3, for each expert, the training loss includes the sample-aware logit adjustment loss in Eq. 4 and regularization loss in Eq. 7. Assuming there are $K$ experts, the overall training loss for PECEL is

$$\mathcal{L}_{all} = \sum_{i=1}^{K}(\mathcal{L}_{sla}(\mathbf{z}^i, \mathbf{y}) + \mathcal{L}_{reg}(\mathbf{P}^i_{down}) + \mathcal{L}_{reg}(\mathbf{P}^i_{up})), \tag{8}$$

where $\mathbf{P}^i_{down}$ and $\mathbf{P}^i_{up}$ denote the projection matrices for expert $i$.

### 3.6 Trainable Parameter Analysis.

To demonstrate the parameter efficiency of the proposed method, we calculate the exact number of trainable parameters when applying AdaptFormer with and without parameter sharing. In practice, the down and up projections are usually instantiated by linear layers, which include the weight matrices and bias vectors. Therefore, an AdaptFormer block has $(2dD + D + d)$ parameters.

**Table 1: Accuracy on CIFAR100-LT under different imbalance ratio (IR) settings. †: lower bottleneck dimension. The best and second-best results are highlighted in bold and underline, respectively.**

| | Backbone | Trainable Params | Total Params | IR=100 | IR=50 | IR=10 |
|---|---|---|---|---|---|---|
| *Training from scratch.* | | | | | | |
| PaCo [7] ICCV'21 | ResNet32 | 0.46M | 0.46M | 52.0 | 56.0 | 64.2 |
| SADE [54] NeurIPS'22 | ResNet32 | 1.34M | 1.34M | 49.8 | 53.9 | 63.6 |
| NCL [27] CVPR'22 | ResNet32 | 1.34M | 1.34M | 54.2 | 58.2 | - |
| BCL [58] CVPR'22 | ResNet32 | 0.46M | 0.46M | 51.9 | 56.6 | 64.9 |
| OUR [35] ACM MM'23 | ResNet32 | 0.46M | 0.46M | 50.8 | 64.5 | 63.9 |
| GLMC [12] CVPR'23 | ResNet32 | 0.46M | 0.46M | 57.1 | 62.3 | 72.3 |
| MDCS [55] ICCV'23 | ResNet32 | 1.34M | 1.34M | 56.1 | 60.1 | - |
| LGLA [45] ICCV'23 | ResNet32 | 1.34M | 1.34M | 57.2 | 61.6 | - |
| DODA [49] ICLR'24 | ResNet32 | 0.46M | 0.46M | 51.0 | 53.6 | 62.7 |
| *Fine-tuning pretrained models.* | | | | | | |
| Zero-Shot | ViT-B | - | - | 64.4 | 64.4 | 64.4 |
| Full Fine-tuning | ViT-B | 85.0M | 85.0M | 52.9 | 61.4 | 73.3 |
| BALLAD [34] arXiv'22 | ViT-B | 149.6M | 149.6M | 77.8 | - | - |
| LiVT [52] CVPR'23 | ViT-B | 85.5M | 85.5M | 58.2 | - | 69.2 |
| LIFT [43] ICML'24 | ViT-B | 0.10M | 85.1M | 81.7 | 83.1 | 84.9 |
| PECEL† | ViT-B | **0.03M** | 85.0M | 81.7 | 83.4 | 85.0 |
| PECEL | ViT-B | 0.10M | 85.1M | **84.3** | **84.6** | **86.4** |

The total number of parameters for $L$ blocks is $(2dD + D + d)L$. Similarly, when using parameter sharing, $L$ AdaptFormer blocks contain $2(d^2 + d)L + (2dD + D + d)$ parameters. Givens $d \ll D$, $2(d^2+d)L+(2dD+D+d)$ is remarkably smaller than $(2dD+D+d)L$. For example, assuming the feature dimension $D$, bottleneck dimension $d$ and the number of AdaptFormer blocks $L$ are 768, 32 and 12, respectively, there are ~599K trainable parameters in all PEFT blocks. When using the proposed parameter sharing strategy, the number of trainable parameters is reduced by ~87% to ~75K.

## 4 Experiments

### 4.1 Experimental Settings

**Datasets and Evaluation**. We conduct experiments on the CIFAR100-LT [26], ImageNet-LT [9], Places-LT [57] and iNaturalist 2018 [47] to verify the efficacy of the proposed method.

**CIFAR100-LT** [26] is the long-tailed version of CIFAR100, including 100 classes with 50K images for training and 10K images for validation. We follow the split in [43, 51, 54] and generate CIFAR100-LT with different imbalance ratios (IR) [45, 55]. IR is defined as the ratio of the maximum and minimum number of samples per class.

**ImageNet-LT** [9] and **Places-LT** [57] are the long-tailed version of ImageNet-1k [9] and Places365 [57] dataset, respectively. ImageNet-LT includes 115.8K training images and 50K validation images, distributed across 1K classes. Places-LT includes 365 classes, with 62.5K training images and 36.5K validation images. The IR of ImageNet-LT and Places-LT are 256 and 996, respectively.

**iNaturalist 2018** [47] consists of 437.5K training images and 244K validation images, distributed across 8192 classes. The maximum and minimum number of samples per class are respectively 1000 and 2, *i.e.*, the IR of iNaturalist 2018 is 500.

By default, we report the Top-1 overall accuracy as the evaluation criterion. Following the common practice [43, 55], we also report

the accuracy of many-shot (with > 100 samples), medium-shot (with $20 - 100$ samples) and few-shot classes (with < 20 samples). Due to the page limits, the results on Places-LT are reported in the supplementary material.

**Implementation Details**. For a fair comparison with previous works [16, 34, 43, 46], we adopt CLIP [40] as the pretrained foundation model. Specifically, the backbone architecture of PECEL is ViT-B [11], and initialized with the pretrained CLIP visual encoder. For parameters in PEFT modules, the up projection layers and learnable scale factors are initialized with 0 to avoid influencing the features at the initial training stage. The other parameters are initialized via Kaiming Normalization [14]. We use the semantic-aware initialization and test-time ensembling proposed in LIFT [43] to initialize the classifiers and further improve the test accuracy. For the experiments on CIFAR100-LT, ImageNet-LT and Place-LT, we use the SGD optimizer to perform optimization. The initial learning rate and total epochs are set as 0.1 and 10. The learning rate decays linearly to 0 by epoch. Since iNaturalist 2018 consists of more training samples and classes, we use the AdamW optimizer [33] for better convergence in experiments on iNaturalist 2018. The initial learning rate and total epochs are set as $5e-4$ and 20, respectively.

Unless specified otherwise, the number of experts in PECEL is set as 3, with $\tau$ in Eq. 4 set as $[0.5, 1.0, 1.5]$ to enforce different experts to learn expertise on the many-shot, medium-shot and few-shot classes, respectively. The bias factor $\epsilon$ in $\beta$ (Eq. 4) is set as 0.1. The bottleneck dimension $d$ in the PEFT blocks is set as 14/64/24/224 for CIFAR100-LT/ImageNet-LT/Places-LT/iNaturalist 2018 to align the number of trainable parameters with LIFT [43]. More implementation details can be found in the supplementary.

## 4.2 Comparison with SOTA Methods

**CIFAR100-LT**. In Table 1, we report the classification accuracy on CIFAR100-LT with different imbalance ratios (IR). † denotes PECEL with a lower bottleneck dimension, *i.e., d* = 1 for CIFAR100-LT. As shown in Table 1, owing to the multiple complementary experts, the proposed PECEL can achieve 84.3%/84.6%/86.4% Top-1 accuracy on CIFAR100-LT with IR=100/50/10, which outperforms previous state-of-the-art LIFT [43] by 2.6%/1.2%/1.4%. Besides, compared to LIFT [43], due to the proposed parameter sharing strategy, the proposed PECEL can achieve comparable performance with ~70% fewer trainable parameters.

**ImageNet-LT & iNaturalist 2018**. The results on ImageNet-LT and iNaturalist 2018 are reported in Table 2. † denotes PECEL with a lower bottleneck dimension, *i.e., d* = 32 for ImageNet-LT and *d* = 128 for iNaturalist 2018. In Table 2, the proposed PECEL achieves 79.2% and 82.5% classification accuracy on ImageNet-LT and iNaturalist 2018, respectively, which remarkably surpass the previous state-of-the-art method LIFT [43]. For the parameter efficiency, compared with LIFT, PECEL can achieve comparable performance with 60.4%/62.3% fewer parameters on ImageNet-LT/iNaturalist 2018, which is attributed to the parameter sharing strategy in Section 3.4. In Table 2, we also report the accuracy of many-shot classes (Many), medium-shot classes (Med) and few-shot classes (Few). As shown in Table 2, due to the proposed skill-diverse and complementary expert learning in Eq. 4, compared with LIFT

[43], PECEL can achieve a more balanced performance across different class groups and higher performance on each class group.

## 4.3 Ablation Study and Analysis

**Ablation Study**. To demonstrate the effectiveness of the proposed modules, we conduct ablation experiments on CIFAR100-LT (IR=100) and iNaturalist 2018 (iNat 2018). The results are reported in Table 3. Our baseline is LIFT [43], which adopts PEFT with AdaptFormer [5] for LTR. In Table 3, we observe that the parameter sharing (Share) strategy can reduce the number of trainable parameters, yet at the cost of reducing the representation capability of all expert models. The proposed parameter regularization (Reg) strategy can reduce representation redundancy and improve the recognition accuracy. Incorporating complementary expert learning (CEL) can substantially improve the overall accuracy of CIFAR100-LT and iNaturalist 2018 to 83.7% and 81.6%, respectively. Moreover, since CEL via sample-aware logit adjustment (CEL-SLA) can facilitate learning on misclassified samples, the final recognition accuracy on CIFAR100-LT and iNaturalist 2018 is further promoted to 84.3% and 82.5%, respectively.

**Complementary Expert Learning**. Table 3 shows the proposed modules can effectively improve the overall accuracy. We need to further verify the contribution of individual experts. In Table 4, we present the classification accuracy of each expert in PECEL on each class group. Table 4 shows that different experts (*i.e.,* $\mathbb{E}_1, \mathbb{E}_2, \mathbb{E}_3$) conditioned with $\tau = [0.5, 1.0, 1.5]$ can competently learn expertise on many-shot, medium-shot and few-shot classes, respectively, which demonstrates the effectiveness of complementary expert learning in Eq. 4. Besides, Table 4 also shows that aggregating these complementary experts yields higher overall accuracy than each single expert, which also verifies the necessity of learning multiple complementary experts.

**Expert Numbers**. As mentioned in Section 4.1, by default, PECEL consists of 3 experts conditioned by the weight factors of 0.5, 1.0 and 1.5, respectively. In Table 5, we report the accuracy of PECEL with different expert numbers and weight factors $\tau$. For example, $[0.5, 1.5]$ denotes PECEL consists of two experts which are conditioned by 0.5 and 1.5, respectively. As shown in Table 5, PECEL with different $\tau$ consistently performs better than LIFT in Table 1. Since the performance of few-shot and many-shot classes will be hindered more remarkably when $\tau = 0.5$ and 2.0, PECEL with $\tau = [0.0, 1.0, 2.0]$ exhibits lower performance than $\tau = [0.5, 1.0, 1.5]$. It's also noted that PECEL achieves 83.9% accuracy when $\tau = [1.0, 1.0, 1.0]$, indicating that ensembling experts with similar expertise can also help.

**Sample-aware Logit Adjustment**. In addition to learning complementary experts, Sample-aware Logit Adjustment (SLA) also aims to shift more attention to the misclassified samples. To verify this, in Figure 4, we present the accuracy gains of each class in CIFAR100-LT when using the sample-aware bias in Eq. 4. Figure 4 shows that applying sample-aware bias can improve the average accuracy by 2.0% on the few-shot classes, which usually involve more misclassified samples due to the lower class frequency. Given the ablation results in Table 3, the proposed SLA can finely achieve the goal of facilitating learning on misclassified samples and improving the overall performance.

**Table 2: Accuracy on ImageNet-LT and iNaturalist 2018. †: lower bottleneck dimension. The best and second-best results are highlighted in bold and underline, respectively.∗ denotes ImageNet-22k pretrained backbone.**

| | ImageNet-LT | | | | | | | iNaturalist 2018 | | | | | | |
| | Backbone | Trainable Params | Total Params | Many | Med | Few | All | Backbone | Trainable Params | Total Params | Many | Med | Few | All |
|---|---|---|---|---|---|---|---|---|---|---|---|---|---|---|
| *Training from scratch.* | | | | | | | | | | | | | | |
| SADE [54] NeurIPS'22 | ResNeXt50 | 66.1M | 66.1M | 66.5 | 57.0 | 43.5 | 58.8 | ResNet50 | 67.6M | 67.6M | - | - | - | 72.9 |
| NCL [27] CVPR'22 | ResNeXt50 | 66.1M | 66.1M | - | - | - | 60.5 | ResNet50 | 67.6M | 67.6M | 72.7 | 75.6 | 74.5 | 74.9 |
| LiVT [52] CVPR'23 | ViT-B | 85.0M | 85.0M | 73.6 | 56.4 | 41.0 | 60.9 | ViT-B | 85.0M | 85.0M | 78.9 | 76.5 | 74.8 | 76.1 |
| OUR [35] ACM MM'23 | ResNeXt50 | 23.0M | 23.0M | 66.5 | 55.7 | 37.9 | 57.2 | ResNet50 | 23.5M | 23.5M | 70.5 | 73.9 | 74.8 | 73.7 |
| MDCS [55] ICCV'23 | ResNeXt50 | 66.1M | 66.1M | 72.6 | 58.1 | 44.3 | 61.8 | ResNet50 | 67.6M | 67.6M | 76.5 | 75.5 | 75.2 | 75.6 |
| LGLA [45] ICCV'23 | ResNeXt50 | 66.1M | 66.1M | - | - | - | 61.1 | ResNet50 | 67.6M | 67.6M | 70.1 | 76.2 | 77.6 | 76.2 |
| DODA [49] ICLR'24 | ResNeXt50 | 66.1M | 66.1M | 66.9 | 54.1 | 37.4 | 56.9 | ResNet50 | 67.6M | 67.6M | 71.2 | 73.2 | 73.4 | 73.7 |
| *Fine-tuning pretrained models.* | | | | | | | | | | | | | | |
| Zero-Shot | ViT-B | - | - | 65.4 | 63.1 | 64.8 | 64.4 | ViT-B | - | - | 7.6 | 4.0 | 3.5 | 4.2 |
| Full Fine-Tuning | ViT-B | 85.0M | 85.0M | 74.6 | 50.6 | 22.0 | 56.0 | ViT-B | 85.0M | 85.0M | 74.2 | 65.3 | 59.1 | 63.8 |
| BALLAD [34] arXiv'21 | ViT-B | 149.6M | 85.0M | 79.1 | 74.5 | 69.8 | 75.7 | - | - | - | - | - | - | - |
| VL-LTR [46] ECCV'22 | ViT-B | 149.6M | 85.0M | **84.5** | 74.6 | 59.3 | 77.2 | ViT-B | 149.6M | 149.6M | - | - | - | 76.8 |
| RAC [32] CVPR'22 | - | - | - | - | - | - | - | ViT-B | 85.0M | 85.0M | 75.9 | 80.5 | 81.1 | 80.2 |
| UDCPG [16] ACM MM'23 | ResNet50 | 23.5M | 23.5M | 76.4 | 69.5 | 60.2 | 70.9 | ResNet50 | 23.5M | 23.5M | - | - | - | 75.2 |
| LPT [10] ICLR'23 | - | - | - | - | - | - | - | ViT-B* | **1.01M** | 86.0M | - | - | 79.3 | 76.1 |
| LIFT [43] ICML'24 | ViT-B | 0.62M | 85.6M | 81.3 | 77.4 | 73.4 | 78.3 | ViT-B | 4.75M | 89.8M | 74.0 | 80.3 | 82.2 | 80.4 |
| PECEL† | ViT-B | **0.24M** | 85.2M | 81.3 | 77.6 | 73.0 | 78.3 | ViT-B | 1.79M | 86.8M | 76.8 | 80.6 | 81.3 | 80.5 |
| PECEL | ViT-B | 0.62M | 85.6M | 82.1 | **77.8** | **76.3** | **79.2** | ViT-B | 4.68M | 89.8M | **80.2** | **82.5** | **82.9** | **82.5** |

**Table 3: Ablation study on CIFAR100-LT with IR=100 and iNaturalist 2018 (iNat 2018). Share: parameter sharing; Reg: parameter regularization; CEL: complementary expert learning via Eq. 3; CEL-SLA: CEL via sample-aware logit adjustment, *i.e.,* Eq. 4; Param: number of trainable parameters in backbone.**

| | Share | Reg | CEL | CEL-SLA | CIFAR100-LT | | iNat 2018 | |
| | | | | | Acc | Param | Acc | Param |
|---|---|---|---|---|---|---|---|---|
| LIFT | | | | | 81.7 | 0.10M | 80.4 | 4.75M |
| | ✓ | | | | 79.9 | 0.03M | 80.1 | 1.99M |
| | ✓ | ✓ | | | 80.7 | 0.03M | 80.4 | 1.99M |
| | ✓ | | ✓ | | 83.7 | 0.10M | 81.6 | 4.68M |
| | ✓ | ✓ | ✓ | | 84.1 | 0.10M | 82.0 | 4.68M |
| PECEL | ✓ | ✓ | | ✓ | 84.3 | 0.10M | 82.5 | 4.68M |

**Table 4: Accuracy of each expert in PECEL on CIFAR100-LT with IR=100 and ImageNet-LT. The best and second-best results are highlighted in bold and underline, respectively.**

| | CIFAR100-LT (IR=100) | | | | ImageNet-LT | | | |
| | Many | Med | Few | All | Many | Med | Few | All |
|---|---|---|---|---|---|---|---|---|
| $\mathbb{E}_1 (\tau_1 = 0.5)$ | **89.6** | 81.4 | 70.9 | 81.1 | **85.0** | 75.0 | 64.0 | 77.3 |
| $\mathbb{E}_2 (\tau_2 = 1.0)$ | 86.4 | 82.8 | 80.7 | 83.4 | 81.3 | 77.3 | 75.4 | 78.5 |
| $\mathbb{E}_3 (\tau_3 = 1.5)$ | 80.8 | 80.4 | **85.3** | 82.0 | 75.1 | 77.6 | **81.3** | 77.2 |
| PECEL | 87.5 | **83.3** | 81.8 | **84.3** | 82.1 | **77.8** | 76.3 | **79.2** |

**Fewer Parameters**. As shown in Table 1 and 2, using the proposed parameter sharing strategy, PECEL can achieve a favorable balance

**Table 5: Impact of different expert numbers and weight factors on CIFAR100-LT (IR=100).**

| | $\tau$ | Many | Med | Few | All |
|---|---|---|---|---|---|
| 2 Experts | [0.5, 1.5] | 87.1 | 82.3 | 80.5 | 83.4 |
| 3 Experts | [1.0, 1.0, 1.0] | 86.9 | **83.4** | 81.3 | 83.9 |
| | [0.0, 1.0, 2.0] | 87.7 | 81.7 | 81.5 | 83.7 |
| | [0.5, 1.0, 1.5] | 87.5 | 83.3 | **81.8** | **84.3** |
| 4 Experts | [0.5, 0.75, 1.0, 1.5] | **87.8** | 83.1 | 80.4 | 83.9 |
| | [0.5, 1.0, 1.25, 1.5] | 86.2 | 82.6 | 81.7 | 83.6 |

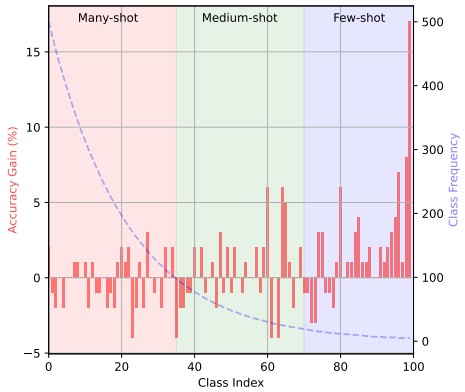

**Figure 4: The accuracy gain of each class in CIFAR100-LT when using Sample-aware Logit Adjustment in Eq. 4.**

between performance and parameter efficiency. To further explore the parameter efficiency for PECEL, we 1) share the projection matrices $\mathbf{P}_{down}$ and $\mathbf{P}_{up}$ in Eq. 5 across different experts (denoted as *Share*); 2) set $\mathbf{P}_{down}$ and $\mathbf{P}_{up}$ as fixed random matrices as done

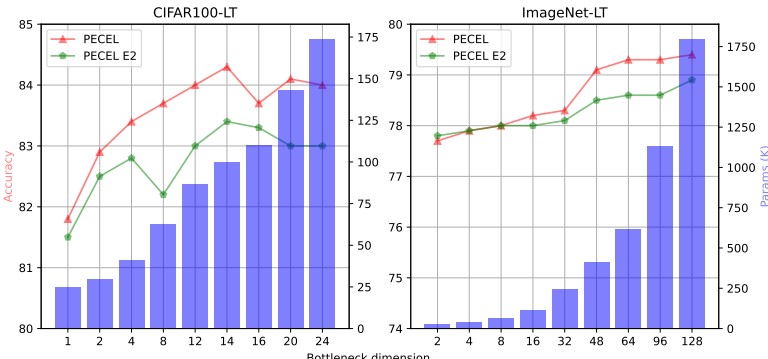

**Figure 5: The accuracy and number of trainable parameters of PECEL with different bottleneck dimensions on CIFAR100-LT and ImageNet-LT. PECEL E2 denotes the expert with $\tau = 1.0$ in PECEL.**

**Table 6: Accuracy on CIFAR100-LT (IR=100) and ImageNet-LT under other settings. ‡: higher bottleneck dimension to align the number of trainable parameters.**

|  | CIFAR100-LT | | ImageNet-LT | |
|---|---|---|---|---|
|  | Param | Acc | Param | Acc |
| LIFT [43] | 0.10M | 81.7 | 0.62M | 78.3 |
| PECEL (*Random*) | 0.04M | 73.6 | 0.32M | 75.8 |
| PECEL (*Share*) | 0.06M | 83.3 | 0.42M | 78.6 |
| PECEL (*Share‡*) | 0.10M | 83.7 | 0.61M | 78.6 |
| PECEL | 0.10M | **84.3** | 0.62M | **79.2** |

**Table 7: Accuracy on CIFAR100-LT (IR=100) and ImageNet-LT with LoRA and ResNet50 as backbone.**

|  | CIFAR100-LT | | ImageNet-LT | |
|---|---|---|---|---|
|  | Param | Acc | Param | Acc |
| LIFT [43] | 0.10M | 81.7 | 0.62M | 78.3 |
| LIFT (LoRA) | 0.15M | 80.8 | 1.18M | 76.9 |
| LIFT (ResNet50) | 0.15M | 63.5 | 0.95M | 70.2 |
| PECEL | 0.10M | 84.3 | 0.62M | 79.2 |
| PECEL (LoRA) | 0.15M | 82.9 | 1.15M | 78.1 |
| PECEL (ResNet50) | 0.16M | 65.9 | 0.88M | 71.2 |

in VeRA [25] (denoted as *Random*). The experimental results on CIFAR100-LT (IR=100) and ImageNet-LT are presented in Table 6. As shown in Table 6, though leveraging 1) and 2) can improve parameter efficiency, the recognition accuracy will also be compromised. The reason is that sharing parameters across experts may confound the learning of skill-diverse experts and the fixed random matrices may weaken the representation learning ability and leave learned representation less discriminative.

**Bottleneck Dimension**. In Figure 5, we present the recognition accuracy and number of trainable parameters of PECEL with different bottleneck dimensions on CIFAR100-LT and ImageNet-LT. As shown in Figure 5, the performance of PECEL generally increases with the number of trainable parameters, *i.e.,* the bottleneck dimension. Considering the overall parameter efficiency, the bottleneck dimensions of CIFAR100-LT and ImageNet-LT are set as 14 and 64, respectively. It's also noted that the performance gains of aggregating different experts (*i.e.,* PECEL E2 to PECEL) also generally enlarge with the increasing bottleneck dimension, which indicates that a larger bottleneck dimension would help learn experts with diverse expertise.

**Generalization Analysis**. The proposed PECEL is instantiated with AdaptFormer [5] and ViT [11], but it's also generalizable to other PEFT methods and backbones. In Table 7, we present the performance of adopting LoRA [20] and ResNet50 [15], respectively. Adopting LoRA for PECEL is straightforward since LoRA also uses the down and up projection matrices. For experiments with ResNet50, each convolutional bottleneck block is equipped

with a parallel PEFT block. Besides, since the latent feature dimension progressively increases in ResNet, in different stages, different projection matrices $P_{down}$ and $P_{up}$ (Eq. 5) with different $D$ are used for parameter sharing. As shown in Table 7, compared with LIFT with the same configuration, the proposed PECEL also performs well and achieves about 2% higher accuracy on CIFAR100-LT, suggesting the proposed PECEL generalizes well to other backbones and PEFT methods.

## 5 Conclusion

In this paper, we propose the Parameter-Efficient Complementary Expert Learning (PECEL) to unleash the potential of pretrained models for LTR via learning diverse complementary experts. To alleviate the extra parameter cost brought by multiple experts, we design a cross-block parameter sharing strategy. Besides, we propose parameter regularization and sample-aware logit adjustment to reduce the shared parameter redundancy and facilitate learning on misclassified samples. The extensive experiments on CIFAR100-LT, ImageNet-LT, Places-LT and iNaturalist 2018 show that the proposed PECEL can achieve comparable performance with previous state-of-the-art methods with about 60% fewer trainable parameters, and achieve substantial improvements with equivalent number of trainable parameters. Currently, the proposed method mainly focuses on leveraging the visual encoder in pretrained models. In the future, we'll explore simultaneously incorporating the visual and linguistic knowledge of pretrained multi-modal foundation models for long-tailed recognition.

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
