# OpenReview forum: "Parameter-Efficient Complementary Expert Learning for Long-Tailed Visual Recognition"
_acmmm.org/ACMMM/2024/Conference — MM2024 Poster_

### Official Review · Reviewer_2pm6 · 2024-05-20

**Rating:** 4
**Confidence:** 4

**Summary:**

This paper addresses the long-tailed recognition (LTR) problem using parameter-efficient fine-tuning (PEFT). It mitigates the bias issue found in previous PEFT long-tailed models (e.g., PEL) by integrating PEFT with complementary experts and a sample-aware logit adjustment loss. Additionally, it reduces computation by sharing some PEFT parameters among experts. Extensive experiments on popular LTR benchmarks demonstrate the effectiveness of the proposed method.

**Strengths:**

This paper has several strengths:

* It is well-written and easy to follow.

* The idea is technically sound, despite its complexity due to multiple components. Fine-tuning from pre-trained models is promising, and PEFT is increasingly popular.

* The results are convincing, as combining diverse experts consistently shows superior performance in LTR.

**Limitations:**

Although this paper is quite readable, some concerns remain:

* The proposed method appears to be a straightforward combination of PEL with diverse experts and LA, both of which are commonly used in LTR, reducing its novelty.

* The motivation for this paper is not clearly stated. While the authors mention that PEL sacrifices performance on tail classes, they do not provide detailed analysis. More empirical evidence would strengthen the paper.

* It is unclear if the topic fits well within the MM community, as the experiments focus on general natural image classification.

Overall, I am somewhat positive about this paper but not fully convinced. I am open to further discussion.

**Suitability:**

2

---

### Official Review · Reviewer_5AHv · 2024-05-22

**Rating:** 4
**Confidence:** 1

**Summary:**

To unleash the intrinsic representation capability of pretrained foundation models, this paper proposes a new Parameter-Efficient Complementary Expert Learn-Asaling (PECEL) for LTR. Specifically, PECEL consists of multiple experts, where individual experts are trained by Parameter-Efficient Fine-Tuning (PEFT) and encouraged to learn different expertise on complementary sub-categories via the proposed sample-aware Zero-Shot logit adjustment loss.

**Strengths:**

1.The proposed PECEL is the first work to effectively learn multiple complementary experts from pretrained foundation models via PEFT for long- tailed recognition.

2.This paper proposes sample-aware logit adjustment and a parameter sharing strategy, which can effectively learn complementary expertise while retaining the parameter efficiency.

**Limitations:**

Please unify the reference format.

**Suitability:**

2

---

### Official Review · Reviewer_YThp · 2024-05-25

**Rating:** 3
**Confidence:** 4

**Summary:**

This work proposes the Parameter-Efficient Complementary Expert Learning method for Long-Tailed Visual Recognition. The PECEL method learns multiple experts from pretrained foundation models via parameter-efficient fine-tuning. The proposed method achieves the best results on all four datasets.

**Strengths:**

1. This work proves that combining parameter-efficient fine-tuning and multiple experts is a feasible solution for long-tailed recognition.

**Limitations:**

1. This work proposes to use multiple experts to address the long-tailed recognition problem. In contrast, previous methods utilizing multiple experts for long-tailed recognition such as RIDE [1], SADE [2], are not mentioned.
2. Following the previous limitation, the proposed model has 85*3+0.1M parameters in total, which is much larger than other SOTAs. In RIDE and SADE, the backbones are shared and the channels are reduced to maintain a comparable total parameter number with previous works. Thus I consider the proposed model has limited concerns about the computation consumption.
3. The second contribution, i.e., sample-aware logit adjustment, has limited contribution from a technical or effect perspective.


[1] Wang, Xudong, et al. "Long-tailed Recognition by Routing Diverse Distribution-Aware Experts." International Conference on Learning Representations. 2020.
[2] Zhang, Yifan, et al. "Self-supervised aggregation of diverse experts for test-agnostic long-tailed recognition." Advances in Neural Information Processing Systems 35 (2022): 34077-34090.

minimal:
L199 UDCPC et al -> UDCPC

**Suitability:**

3

---

### Official Review · Reviewer_UJgZ · 2024-06-07

**Rating:** 4
**Confidence:** 4

**Summary:**

The paper proposes a novel method called Parameter-Efficient Complementary Expert Learning (PECEL) to improve long-tailed recognition (LTR) tasks. Long-tailed recognition deals with datasets where a few classes (head classes) have many samples while many classes (tail classes) have few samples. PECEL addresses this by using multiple experts, each fine-tuned to different subsets of classes using Parameter-Efficient Fine-Tuning (PEFT). The experts' predictions are aggregated to achieve balanced performance across all classes. PECEL also introduces a parameter-sharing strategy to maintain efficiency by reducing the number of trainable parameters. The method shows state-of-the-art performance on several benchmarks.

**Strengths:**

Technical Approach: The paper's technical approach is sound, with clear explanations of how PECEL works and the innovations introduced, such as the sample-aware logit adjustment loss and parameter-sharing strategy.
Adequate Evaluation: Extensive experiments on four LTR benchmarks (CIFAR100-LT, ImageNet-LT, Places-LT, iNaturalist 2018) demonstrate the effectiveness of PECEL, showing superior performance compared to existing methods.
Clarity: The paper is well-organized and clearly written, making the complex methodology accessible and understandable.

**Limitations:**

Lack of Novelty: The proposed method appears to be a combination of existing approaches rather than a truly novel solution. For example, the idea of using multiple experts has already been applied to long-tailed visual recognition in NCL [27], and logit adjustment is a common method in this field [1,2]. The authors need to more clearly articulate the unique contributions and novelty of their approach.

[1] Menon, Aditya Krishna, et al. "Long-tail learning via logit adjustment." arXiv preprint arXiv:2007.07314 (2020).
[2] Zhao, Yan, et al. "Adaptive logit adjustment loss for long-tailed visual recognition." Proceedings of the AAAI conference on artificial intelligence. Vol. 36. No. 3. 2022.
Unclear Comparisons: The paper compares training from scratch with fine-tuning pretrained models, but this comparison lacks significance due to the different datasets and backbone networks used. The superiority of fine-tuning pretrained models over training from scratch is well-established and thus does not provide new insights.

Inadequate Parameter Calculation: The method for calculating the number of parameters seems flawed. The comparison in the paper only considers the parameters adjusted by the authors, ignoring the parameters of the pretrained network. This approach can mislead the evaluation of the actual parameter efficiency and computational requirements of the proposed method.

In conclusion, while the authors' work is valuable, it does not meet the standards required for this conference.

**Suitability:**

2

---

### Meta-Review · Area_Chair_pxbn · 2024-07-01

**Recommendation:** Accept (Poster)
**Confidence:** 5

**Metareview:**

Three of the reviewers are positive on this paper, and one reviewer recommends rejection. The proposed method is reasonable and sound. It is the first work to effectively learn multiple complementary experts from pre-trained foundation models. There are also some issues in this paper, and most of them have been replied in the rebuttal. AC recommends acceptance. Authors need to improve the final version following reviewers' suggestions.